# Perceived Social Support, Depressive Symptoms, Self-Compassion, and Mobile Phone Addiction: A Moderated Mediation Analysis

**DOI:** 10.3390/bs13090769

**Published:** 2023-09-14

**Authors:** Xiaofan Yang, Hang Ma, Ling Zhang, Jinyang Xue, Ping Hu

**Affiliations:** 1Department of Psychology, Renmin University of China, No. 59 Zhongguancun Street, Haidian District, Beijing 100872, China; yxfpsy2022@tjcu.edu.cn (X.Y.); 2021102341@ruc.edu.cn (H.M.); 2018000488@ruc.edu.cn (L.Z.); 2019100823@ruc.edu.cn (J.X.); 2Department of Psychology, Tianjin University of Commerce, No. 409 Guangrong Road, Beichen District, Tianjin 300134, China

**Keywords:** perceived social support, depressive symptoms, self-compassion, mobile phone addiction, college students

## Abstract

Objective: The primary objective of this study is to investigate the relationships between perceived social support and mobile phone addiction, as well as the mediating effect of depressive symptoms and the moderating effect of self-compassion. Methods: A total of 874 college students completed questionnaires, including the perceived social support scale, depression–anxiety–stress scale, mobile phone addiction index, and the short form of the self-compassion scale. The participants included 202 males and 672 females, with an average age of 19.54 (SD = 2.16). Results: A moderated mediation analysis was conducted. The results revealed that perceived social support fully mediated the negative relationship between perceived social support and mobile phone addiction. Self-compassion attenuated the mediating effects. Conclusions: The present study indicated that insufficient perceived social support may increase the risk of mobile phone addiction among college students because of the impact of depressive symptoms. However, self-compassion could buffer this adverse effect.

## 1. Introduction

In today’s increasingly interconnected world, mobile phones are essential mobile internet terminals and have become indispensable in people’s lives [1]. These devices facilitate communication without restrictions related to physical proximity or location and allow users to engage in various online activities, such as online chatting, virtual meetings, online gaming, and other digital services. With the continuous development and expansion of their functionality, mobile phones bring significant convenience to individuals’ social lives and a host of challenges [2]. A growing number of users find it hard to break free from their dependence on mobile phones, making this behavioral addiction a critical issue faced by the whole society [1,3].

To date, mobile phone addiction remains a debated term. Researchers have primarily categorized it into two groups: non-addiction and addiction oriented. Some researchers have conceptualized it as compulsive usage of mobile phones, referring to behaviors, such as the need to carry phones at all times and check them frequently in vital situations like social occasions. This repetitive, compulsive behavior can harm individuals’ social and personal life [4]. For instance, Tayana and Xavier (2018) suggested that researchers should avoid the addiction framework when examining technology-related behaviors [5]. Specifically, they recommended using alternative terms, such as “problematic use”, for a more accurate and nuanced description of these phenomena. However, a larger number of researchers, drawing on related concepts such as internet addiction, argue that excessive mobile phone usage should be categorized as a behavioral and technological addiction, ultimately defining it as mobile phone addiction. This term describes the psychological dependence resulting from individuals’ compulsive use of mobile phones, which results in a loss of control over phone usage and related services, and consequently, it interferes with people’s daily lives, giving rise to psychological or behavioral problems [6,7,8,9].

College students represent a group with a high prevalence of phone addiction [10]. Previous research has demonstrated that mobile phone addiction has widespread and severe adverse effects on college students’ academic performance and physical and mental well-being [8,11]. Therefore, enhancing our comprehension of the factors contributing to mobile phone addiction in college students holds considerable practical importance. To that end, the present research explores the interpersonal and psychological factors that play a role in mobile phone addiction while highlighting potential protective factors that could alleviate the formation of this addiction. The findings of this study could provide valuable insights for efficiently preventing and intervening in mobile phone addiction among college students.

### 1.1. Perceived Social Support and Mobile Phone Addiction

Previous studies indicated that numerous factors contribute to the excessive usage of mobile phones, with the primary one being the lack of perceived social support [12]. As social animals, human beings are inclined to establish positive connections with others and obtain social support in order to foster social development. Relational needs represent one of the fundamental psychological needs of human beings. To some extent, perceived social support reflects how well individuals’ relational needs are met. When individuals cannot feel sufficient social support in real life, such as respect or help from significant others, they may be inclined to compensate for their relationship needs in some way [13]. Mobile phones, serving as communication tools with powerful social functions and high accessibility, have become a practical means for individuals to engage in compensatory social communication. Through mobile phone communication, individuals can maintain suitable interpersonal relationships and receive support from others. According to the compensatory internet use theory, an absence of real-life social support may lead individuals to experience unmet social needs, consequently encouraging alternative online social engagement as a form of compensation [14]. If this compensatory behavior spirals out of control, it can potentially culminate in mobile phone addiction.

The existing research regarding the association between social support and mobile phone addiction has yet to reach definitive conclusions. The poor-get-richer model suggests that people with insufficient perceived social support might take advantage of novel communication channels, like the internet, to forge connections and secure support—a behavior that could potentially result in mobile phone addiction [15,16]. Conversely, the rich-get-richer model proposes that people endowed with abundant social support resources might be more inclined to broaden their social network online, demonstrating online social tendencies that correspond to their existing levels of social support [17]. Furthermore, cross-lagged research has revealed that the lower the level of social support, the higher the risk of mobile phone addiction [18]. Zhao et al. (2021) further distinguished perceived and online social support, finding that perceived social support decreased while online social support, conversely, heightened the risk of mobile phone addiction.

While these findings shed light on this topic, the mediating and moderating factors involved in this association still need to be explored. Consequently, this study attempts to explore more deeply the mediating role of negative emotions and the possible positive buffering variables in order to offer a basis for the prevention and intervention of mobile phone addiction among college students.

### 1.2. The Mediating Role of Depressive Symptoms

Depressive symptoms are common negative emotions that exist in contemporary college students, and the depressive symptoms can significantly affect students’ academic performance and overall well-being. According to cognitive-behavioral theory [19], individuals with depressive symptoms may engage in maladaptive behaviors like excessive mobile phone use as an attempt to alleviate or escape from their negative emotions. In this sense, depressive symptoms can serve as a critical psychological mechanism or “proximal factor” that triggers the addictive behavior. In line with the interpersonal model, social support strongly correlates with depressive symptoms, and individuals who perceive a lower level of social support are more prone to experiencing depressive symptoms [20]. Compensatory internet use theory (CIUT) [14] points to various motivations behind excessive or problematic technology use, with the central issue being the individual’s reaction to adverse events. As a result, people with low perceived social support might experience heightened affective psychopathology symptoms, such as depressive symptoms, which may exacerbate their compensatory behavior—for example, using mobile phones to relieve stress and regulate negative emotions [21]. Consequently, they may increasingly use mobile phones to pursue social connections and alleviate unpleasant feelings [22,23,24]. Additionally, depressive symptoms can also reduce an individual’s self-control, making it harder to resist instant gratification and leading to out-of-control compensatory behavior and eventually developing into mobile phone addiction.

### 1.3. The Moderating Role of Self-Compassion

Previous research primarily investigated the negative factors contributing to the development of mobile phone addiction, while less emphasis was placed on the buffering effect of positive aspects. Several studies have addressed the impacts of positive variables on mobile phone addiction, such as self-control and self-esteem [2,10]. In the realm of positive psychology, self-compassion has recently gained significant attention and is defined as the capacity to treat oneself with kindness when confronted with failures, inadequacies, or distress [25].

Stanton et al. (2000) proposed an emotion-oriented coping strategy as a positive psychological adjustment strategy in which individuals continuously try to explore and understand their emotions to maintain attention [26]. In specific ways, self-compassion is an effective emotionally oriented coping strategy. Self-compassion requires individuals to maintain awareness of and engagement to their emotions and to remain conscious of and understand the meaning of painful emotions. Individuals with high self-compassion would allow emotions to exist, neither ignoring nor denying them, and take appropriate and effective action to transform depressive symptoms into more positive psychological states. Therefore, it may thus reduce the risk of maladaptive behaviors such as mobile phone addiction.

Evidence has demonstrated that self-compassion could alleviate psychological and pathological symptoms, such as depression and anxiety, while improving physical health [27,28,29]. Concerning its underlying mechanisms, self-compassion brings about positive outcomes by reducing maladaptive emotion regulation strategies and fostering adaptive ones [30]. Specifically, when external support is inadequate, individuals with high self-compassion might actively seek internal comfort rather than mindlessly ruminating over isolation, making negative internal attributions and self-blame. This could compensate for the consequences of insufficient social relationships, curtail the development of depressive symptoms, and ultimately reduce the likelihood of mobile phone addiction.

### 1.4. The Present Study

The present study aims to examine the underlying mediator (depressive symptoms) and moderator (self-compassion) between perceived social support and mobile phone addiction among college students. Specifically, this study constructed a moderated mediation model to test three main hypotheses:

**Hypothesis** **1.**
*Depressive symptoms mediate the relationship between perceived social support and mobile phone addiction.*


**Hypothesis** **2a.**
*Self-compassion moderates the direct association. Perceived social support has a weaker association with mobile phone addiction among college students with a high level of self-compassion than those with a low level of self-compassion.*


**Hypothesis** **2b.**
*Self-compassion moderates the mediation processes. Perceived social support has weaker associations with depressive symptoms, which in turn have weaker associations with mobile phone addiction among college students with a high level of self-compassion compared to those with a low level of self-compassion.*


The theoretical model is outlined in Figure 1.

## 2. Methods

### 2.1. Participants and Procedure

Data collection for this research involved circulating an online questionnaire among college students in Hebei Province, China. In total, 986 questionnaires were disseminated via an online survey-collection platform. After data cleaning, 874 questionnaires were determined valid, with an effective response rate of 88.64%. The participants included 202 males and 672 females, with an average age of 19.54 (SD = 2.16). The sample distribution among class years was 437 freshmen, 211 sophomores, 110 juniors, and 116 seniors.

The procedure and content of this study conformed to the ethical standards established by the Ethics Committee of The Department of Psychology, Renmin University of China (ethical approval number: 21-029).

### 2.2. Measures

#### 2.2.1. Perceived Social Support Scale

The perceived social support scale encompasses three dimensions: support from parents, teachers, and friends [31]. It consists of 11 items, and each item is scored on a 5-point Likert scale (0 = strongly disagree, 5 = strongly agree). The sum of the scores represents the individual’s subjective perception of social support, with higher scores signifying a stronger sense of perceived social support. The Cronbach’s α coefficient of the scale in this study was 0.913.

#### 2.2.2. Mobile Phone Addiction Index (MPAI)

Mobile phone addiction was evaluated using the mobile phone addiction index [32], which features 17 items rated on a 5-point Likert scale (1 = never, 5 = always). It includes four dimensions: loss of control, withdrawal, avoidance, and inefficiency. A composite score was derived by averaging all the items. Higher scores indicate higher mobile phone dependency. In the present study, the Cronbach’s α coefficient for this scale was 0.898.

#### 2.2.3. Depression–Anxiety–Stress Scale

Depressive symptoms were assessed using the depression subscale of DASS-21 [33]. It consisted of 21 items measuring depression, anxiety and stress. Seven items specifically targeting depressive symptoms. The Likert 4-point score (0 = agree, 3 = always agree) was adopted. For the main thrust of our investigation, we focused solely on the depression dimension. The Cronbach’s α coefficient of this scale in this study was 0.911.

#### 2.2.4. Self-Compassion Scale-Short Form

This study used the short form self-compassion scale (SCS-SF) [34] to measure individual self-compassion. The brief scale consists of 12 items divided into three dimensions: self-kindness, common humanity, and mindfulness. Participants scored their responses on a 5-point Likert scale (0 = completely disagree, 5 = completely agree). After reverse coding certain items, higher overall scores represented a greater degree of self-compassion. The Cronbach’s α coefficient of this scale in this study was 0.822.

### 2.3. Data Processing

SPSS 24.0 (IBM, Armonk, NY, USA) was utilized for the common method bias test and correlation analysis, while the SPSS-PROCESS plugin was employed to examine the mediating effects and conduct moderated mediation model tests.

## 3. Results

### 3.1. Common Method Bias Test

As all the variables were evaluated through self-report questionnaires, this might lead to an issue of common-method bias. Harman’s single-factor test was employed to address common-method bias. The results revealed that 10 factors have eigenvalues greater than 1. The variance accounted for by the first factor was 23.50%, falling short of the critical threshold of 40%. Consequently, no severe common-method bias problem was evident in this study.

### 3.2. Descriptive Analysis

Perceived social support negatively correlated with depressive symptoms and mobile phone addiction. A significant positive correlation was observed between depressive symptoms and mobile phone addiction. Self-compassion showed a significant positive relationship with perceived social support, while being significantly negatively correlated with mobile phone addiction and depressive symptoms. See Table 1.

### 3.3. Perceived Social Support and Mobile Phone Addiction: A Moderated Mediation Model Test

First, model 4 in the SPSS-PROCESS plugin was utilized to examine the mediation effect. Based on the existing research [35,36,37,38], to avoid any potential confusion and to focus on our main objective, we have taken gender, age, and grade as control variables. The results indicated that perceived social support significantly and negatively predicted college students’ mobile phone addiction (*β* = −0.17, *t* = −5.19, *p* < 0.001). After including depressive symptoms as a mediator variable, perceived social support significantly negatively predicted depressive symptoms (*β* = −0.46, *t* = −15.44, *p* < 0.001), and depressive symptoms significantly positively predicted mobile phone addiction (*β* = 0.45, *t* = 12.72, *p* < 0.001). In contrast, perceived social support no longer predicted mobile phone addiction (*β* = 0.03, *t* = 0.90, *p* = 0.37), as shown in Table 2. The bootstrap test showed that the mediating effect of depressive symptoms was significant, the effect value was −0.21, the 95% confidence interval was [−0.24, −0.17], and the mediating effect accounted for 86.79% of the total effect. These results suggest that depressive symptoms have a completely mediating role between perceived social support and mobile phone addiction.

Second, a moderated mediation model test was performed using the SPSS PROCESS macro, Model 59. The bootstrap test was used, and sampling was repeated 5000 times. The results are shown in Table 3 and Figure 2 and Figure 3, which demonstrated that when self-compassion was added as a moderator, the interaction terms of perceived social support and self-compassion significantly positively predicted depressive symptoms (*β* = 0.08, *t* = 3.25, *p* < 0.01). Simple slope analysis revealed that perceived social support significantly negatively predicted depressive symptoms when self-compassion was low (M − 1SD) (simple slope = −0.42, *t* = −10.85, *p* < 0.001), while when self-compassion was high (M + 1SD), the association between perceived social support and depressive symptoms was attenuated (simple slope = −0.26, *t* = −6.72, *p* < 0.001), which demonstrated that individuals with higher self-compassion experienced fewer depressive symptoms when perceived social support declined.

In addition, the interaction terms of perceived social support and self-compassion significantly negatively predicted mobile phone addiction (β = −0.09, *t* = −2.69, *p* < 0.01). Simple slope analysis revealed that perceived social support was significantly positively correlated with mobile phone addiction when self-compassion was low (M − 1SD) (simple slope = 0.17, *t* = 3.49, *p* < 0.01). In contrast, when self-compassion was high (M + 1SD), this association became nonsignificant (simple slope = −0.01, *t* = −0.27, *p* = 0.79), which demonstrated that for those with lower self-compassion, the direct path of perceived social support and mobile phone addiction was covariant in a positive manner.

Lastly, the mediating effect of depressive symptoms was tested at different levels of self-compassion. The results are shown in Table 4. At low self-compassion levels, the mediating effect was −0.16 with a 95% confidence interval of (−0.22, −0.12); when self-compassion was high, the mediating effect was −0.13 with a 95% confidence interval of (−0.18, −0.09). The results indicated that with the increase in self-compassion, the decrease in perceived social support will lead to less mobile phone addiction through depressive symptoms. In other words, our moderated mediation analysis revealed that self-compassion mitigated the indirect impact of perceived social support on mobile phone addiction via depressive symptoms. This finding aligns with our expectations and represents a pioneering discovery in the field.

## 4. Discussion

The rapid development of mobile internet has positioned mobile phones as a double-edged sword. Therefore, understanding the formation process of mobile phone addiction and identifying effective prevention and control measures have become focal points for academia worldwide.

Many studies have explored the influencing factors of mobile phone addiction, including environmental, social, and psychological factors. Some scholars believe that addictive behavior cannot be explained by a single factor but is instead the result of a combination of factors [39]. Previous studies have explored the moderating role of self-compassion in various contexts, such as its impact on the relationship between stress and relevant behavioral outcomes [40,41]. While these studies have demonstrated the protective effects of self-compassion on adverse psychological consequences, they have rarely focused on mobile phone addiction, particularly in a university student population. Our study is the first to explore the moderating role of self-compassion in the relationship between perceived social support and mobile phone addiction. The findings suggest that inadequate perceived social support increased the risk of mobile phone addiction among college students because of the role of depressive symptoms. Nevertheless, self-compassion could buffer this adverse effect.

### 4.1. Perceived Social Support and Mobile Phone Addiction

In this research, the findings supported the majority of the hypotheses. First, results showed that perceived social support negatively predicted mobile phone addiction. Individuals with low social support have a higher risk of mobile phone addiction, consistent with previous studies [16]. According to compensatory internet use, individuals may turn to the mobile network to obtain support from socially distanced individuals, such as faraway parents, friends, net friends, etc. In addition, to escape social frustration in real life, individuals are more likely to use the internet as a distraction. For instance, those who merely receive social support are more likely to spend time on mobile phones to mitigate negative emotional feelings in real life.

### 4.2. The Mediating Role of Depressive Symptom

Numerous studies have pointed out that depressive symptoms are risk factors for mobile phone addiction [22,24,42]. This research further confirmed that depressive symptoms fully mediated the relationship between perceived social support and mobile phone addiction, corroborating that emotional factors significantly influence addictive behaviors.

According to general strain theory [43], lack of perceived social support could be seen as a stressor that often leads to adverse emotional experiences (such as depressive symptoms), and in order to cope with this stress and alleviate the adverse emotional experiences, individuals often seek a short and quick way for compensation. In addition, as an essential source of information for self-evaluation, perceived social support would affect an individual’s self-value judgment [44]. Especially in a collectivist culture, the attitude of significant others will largely influence an individual’s view of the self. Lack of perceived social support will lead to negative self-perceptions [44], reduce the sense of self-value, and bring depressive symptoms. In addition, individuals with depressive symptoms might use mobile phones more to relieve their negative feelings [45]. Persistent depression could potentially lead individuals to retreat from reality habitually, seeking solace in the virtual world instead. This continual reliance on the online sphere might result in a dependence on mobile phones, establishing a concerning pattern over time [24,46].

In terms of the characteristics of the device, rich functions such as mobile phone games, online shopping, and short video platforms can provide individuals with instant gratification anytime and anywhere, bringing sensory pleasure and becoming a haven for college students to cope with the loss of reality and relieve negative emotions. At the same time, the negative mood accompanying depressive symptoms will weaken the individual’s executive and inhibitory function [47], thus making it more difficult for individuals to resist temptation and indulge in the short-term pleasure brought by mobile phones.

Moreover, results showed that when depressive symptoms are included as a mediating variable, the negative predictive effect of perceived social support on mobile phone addiction is no longer significant, indicating that it is the subjective emotional experience of individuals that determines whether compensatory behavior happens. According to the compensatory theory, whether compensation is excessive and out of control depends on how individuals evaluate adverse situations. When social needs cannot be met, the depressive symptoms and other negative moods that come with them may be the direct cause of mobile phone addiction. In other words, it is possible that the lack of perceived social support does not necessarily lead to mobile phone addiction. What is critical is the individual’s emotional experience in the face of low social support.

### 4.3. The Moderating Role of Self-Compassion

According to the stress buffer hypothesis, favorable individual factors can mitigate the effect of stress on a person’s bad mental and behavioral outcomes [48]. This study suggests that self-compassion, one of the positive psychological qualities, could buffer the adverse effects of interpersonal stress, i.e., lack of social resources on these outcomes. Specifically, self-compassion moderated the path of perceived social support–mobile phone addiction and perceived social support–depressive symptoms, which supports H2a and H2b.

It has been found that self-compassion is not only negatively correlated with depressive symptoms but can also act as a buffer against depressive symptoms caused by environmental stress [49]. That is, individual factors weakened the adverse effects of external risk factors [44]. In the absence of external support, college students with high levels of self-compassion can gain strength from themselves and view their situation from a broader and objective perspective, thereby weakening the impact of negative social feelings and promoting a positive self-adjustment [28,49]. This will effectively prevent the onset of depressive symptoms, help to maintain positive physical and mental states, and thus reduce maladaptive behaviors such as mobile phone addiction.

What is more, results indicated that self-compassion moderates the direct path from social support to mobile phone addiction. Specifically, for those with low self-compassion, perceived social support increased the risk of mobile phone addiction, while for those with high self-compassion, the relation between them became nonsignificant, indicating that those who are compassionate to themselves might not turn to mobile phone for compensation when they can hardly get social support. This may be because individuals with high self-compassion are likely to be less affected by the amount of social support. Therefore, self-compassion could effectively break the typical response to low social support, mitigating the potential risk of mobile phone addiction. These findings underline the importance of fostering self-compassion to counteract the negative impact of inadequate social support.

While our results indicate a nonsignificant moderating effect of self-compassion between depressive symptoms and mobile phone addiction, this does not necessarily imply its insignificance in the broader context of mental health and behavior. As a psychological trait, self-compassion might play a more preventative role by possibly reducing the onset of depressive symptoms rather than mitigating the behavior (excessive mobile phone usage) that may occur once depressive symptoms have manifested. To put it differently, individuals with high levels of self-compassion might generally be less prone to develop depressive symptoms, which in turn reduces the possibility of engaging in addictive mobile phone behavior.

### 4.4. Theoretical Implications

First, our research broadens the understanding of mobile phone addiction among college students. We have analyzed this within an integrated framework incorporating perceived social support, depressive symptoms, and self-compassion. Consequently, our study not only supports but also extends the compensatory internet use theory [14] and the interpersonal model [48].

Second, our research introduces the concept of self-compassion as a moderating variable in this relationship, which is a relatively newer area in the realm of positive psychology pertaining to addictive behaviors. Our findings underscore the potential of positive psychological constructs, e.g., self-compassion, as instrumental in driving positive behavior change even in adverse situations. This cultivates a more nuanced understanding of protective factors in the onset of mobile phone addiction and aligns with the principles of an emotion-oriented coping strategy [26].

Finally, in practice, our research sparks reconsideration of intervention strategies addressing addiction. Moving from traditional external support systems, it presents the promise of harnessing internal resourcefulness by cultivating self-compassion. This inspiration can potentially develop into a new theoretical framework for addressing addiction and similar maladaptive behavior patterns.

### 4.5. Implications for Prevention and Intervention

Based on the results of this study, several suggestions are raised that may provide certain references for the prevention and intervention of mobile phone addiction in college students.

First, parents, colleges, and relevant organizations should provide adequate social support to college students. It is also important to cultivate students’ social skills and gratitude abilities to make the most of the external social support to avoid out-of-control compensation behavior.

Second, since depressive symptoms can exacerbate maladaptive phone use, measures to prevent depressive symptoms will be conducive to reducing the risk of phone addiction. For example, mental health teachers can guide college students to establish positive self-concept and teach emotional regulation skills to reduce the occurrence of depressive symptoms.

Last, considering that self-compassion plays an important role in buffering the adverse effect of social risk factors on mobile phone addiction, it is very necessary for mental health departments and educators to guide and train college students to learn positive emotional coping strategies and foster self-compassion. As self-compassion training has been demonstrated as an effective intervention technique [50], more of such interventions can be carried out among college students to prevent and reduce the risk of mobile phone addiction.

### 4.6. Limitations and Future Direction

There are still limitations in this study, which can be improved and explored in future research. First, this study is a cross-sectional study, so we must be cautious in inferring causal relationships between these variables. Studies have found that mobile phone addiction can also lead to depression, anxiety, and stress [11,21], so future studies could use longitudinal or experimental designs to verify the causal relationship further.

Second, this study was based on self-reported data from online questionnaires. Research indicates that self-reported mobile phone usage does not consistently correlate strongly with objectively measured mobile phone use [51,52]. Thus, future studies could adopt the multi-source method to collect data. What is more, the gender proportion of participants is unbalanced. Although this study uses gender as a covariant in data analysis, it might affect the generalization of the results.

Third, this study found that depressive symptoms could mediate perceived social support’s effect on mobile phone addiction. However, whether other negative emotions (such as anxiety, etc.) might have the same impact remains to be further explored.

Fourth, our research focuses only on the moderating role of one positive psychological variable—self-compassion. However, many other positive psychological variables such as self-efficacy, optimism, and resilience may play a similar role. Future research might further investigate the interaction between these variables and their cumulative effects on individuals’ responses to adverse situations.

Finally, breaking down the MPAI into its four dimensions (i.e., loss of control, withdrawal, avoidance, and inefficiency) could provide valuable insights into the specific aspects of mobile phone addiction that are most influenced by perceived social support, self-compassion, and depressive symptoms. Considering this, we see a valuable direction for future research in carrying out a more elaborate separate dimension analysis of the MPAI. In addition, it is still unclear which functions of mobile phones (entertainment, information seeking, learning, and so on) are related to the lack of social support, which is also worthy of exploration in future research.

## 5. Conclusions

The findings provide new insights into the relationship between perceived social support, depressive symptoms, self-compassion, and mobile phone addiction and how these factors work together to influence mobile phone addiction. In conclusion, the current study provides a possible explanatory mechanism and empirical support for the mobile phone addiction of college students.

## Figures and Tables

**Figure 1 behavsci-13-00769-f001:**
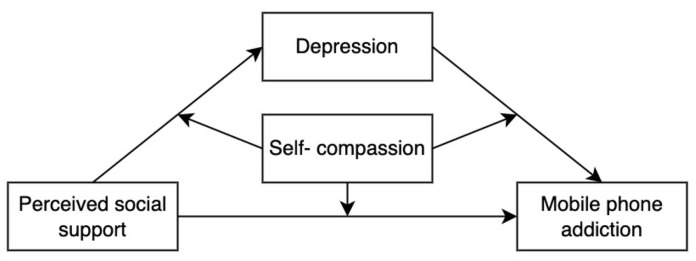
The theoretical model of this study.

**Figure 2 behavsci-13-00769-f002:**
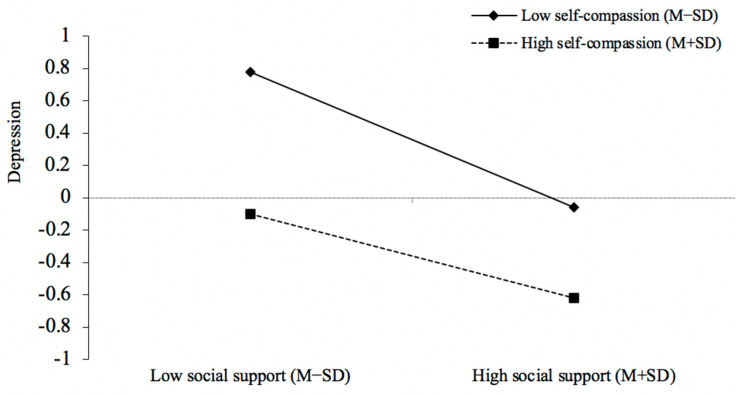
The simple slope analysis (perceived social support–depressive symptoms). In our sample size of (874), 227 fell within M + 1SD and 443 within M − 1SD for the self-compassion measure.

**Figure 3 behavsci-13-00769-f003:**
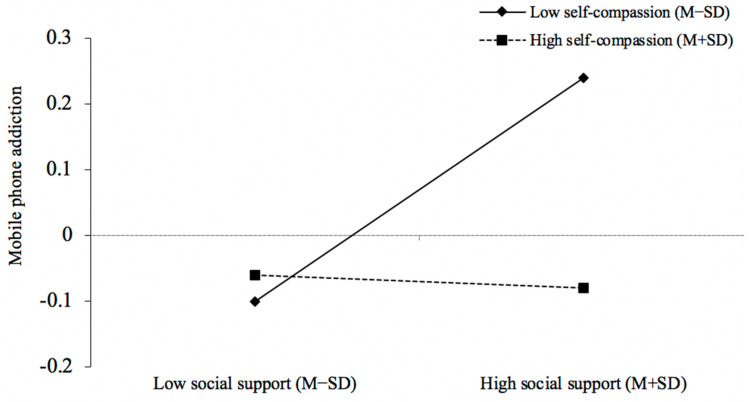
The simple slope analysis (perceived social support–mobile phone addiction). In our sample size of (874), 227 fell within M + 1SD and 443 within M − 1SD for the self-compassion measure.

**Table 1 behavsci-13-00769-t001:** Correlations among the studied variables (*N* = 874).

	M ± SD	1	2	3	4
1. Perceived social support	3.55 ± 0.81	1			
2. Mobile phone addiction	2.80 ± 0.73	−0.18 ***	1		
3. Depressive symptoms	1.82 ± 0.68	−0.48 ***	0.43 ***	1	
4. Self-compassion	3.12 ± 0.43	0.39 ***	−0.29 ***	−0.46 ***	1

Note: *** *p* < 0.001.

**Table 2 behavsci-13-00769-t002:** The mediating effect test of depressive symptoms.

Regression Equation (*N* = 874)		Fit Metrics			Coefficient Significance	
Outcome Variable	Predictor Variable	*R*	*R^2^*	*F*	*β*	*t*
Depressive symptoms		0.49	0.24	69.22 ***		
	Perceived social support				−0.46	−15.44 ***
Mobile phone addiction		0.56	0.31	77.40 ***		
	Depressive symptoms				0.45	12.72 ***
	Perceived social support				0.03	0.90

Note: All variables were standardized and brought into the regression equation; *** *p* < 0.001.

**Table 3 behavsci-13-00769-t003:** Moderated mediation model test.

Predictors	Model 1		Model 2		Model 3	
	(MPA)		(Dep)		(MPA)	
	*β*	*t*	*β*	*t*	*β*	*t*
PSS	−0.07	−2.05 *	−0.34	−11.28 ***	0.08	2.19 *
S-C	−0.22	−6.06 ***	−0.36	−11.38 ***	−0.07	−1.92
PSS × S-C	−0.09	−3.06 *	0.08	3.25 **	−0.09	−2.69 **
Dep					0.44	11.47 ***
Dep × S-C					0.05	1.58
*R*	0.31		0.58		0.47	
*R^2^*	0.10		0.34		0.22	
*F*	23.20 ***		110.35 ***		40.06 ***	

*N* = 874. The beta values are standardized coefficients. Gender is encoded as dummy variable and controlled in the model. PSS = perceived social support, S-C = self-compassion, Dep = depressive symptoms, MPA = mobile phone addiction. * *p* < 0.05, ** *p* < 0.01, *** *p* < 0.001.

**Table 4 behavsci-13-00769-t004:** Mediation model test.

Predictors	Model 1		Model 2		Model 3	
	(MPA)		(Dep)		(MPA)	
	*β*	*t*	*β*	*t*	*β*	*t*
PSS	−0.18	−5.28 ***	−0.46	−15.44 ***	0.03	0.90
Dep					0.45	12.72 ***
*R*	0.18		0.49		0.56	
*R^2^*	0.03		0.24		0.31	
*F*	14.28 ***		69.22 ***		77.40 ***	

All variables were standardized in the model; PSS = perceived social support, Dep = depressive symptoms, MPA = mobile phone addiction. *** *p* < 0.001.

## Data Availability

The authors will make the raw data that supports the conclusions of this article available without any undue reservation.

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
