# Peer review of "Perceived Social Support, Depressive Symptoms, Self-Compassion, and Mobile Phone Addiction: A Moderated Mediation Analysis"

_behavsci, 2023, doi:10.3390/bs13090769_

Round 1
Reviewer 1 Report
This is an interesting study investigation the moderating effect of self-compassion on the relationship between social support, depression and mobile phone use. There are many issues of grammar/spelling in the paper.
The section on the mediation moderation analysis is a little confusing – why is depression the outcome variable in the one analysis? From what I had read, smartphone usage was the outcome variable. I found the results section confusing, and not always clear.
The discussion starts of well explaining why depression may predict mobile phone use. However, the discussion surrounding the mediating and moderating effects is less clear and repetitive.
I have left several comments in the attached PDF.

There are several places where the spelling and grammar is incorrect. I suggested a proofreader be employed to fix these errors as it often made the article hard to read.
Author Response
- This is an interesting study investigation the moderating effect of self-compassion on the relationship between social support, depression and mobile phone use. There are many issues of grammar/spelling in the paper.
[Response]: We would like to convey our gratitude for the time you took in reviewing our manuscript and providing us with valuable feedback. We acknowledge the grammatical and spelling errors pointed out by you and sincerely apologize for these oversights. We have acted promptly and diligently by engaging a a colleague fluent in English writing to ensure the paper reads coherently and accurately.We hope that the revised manuscript now more effectively communicates the results of our research.
- The section on the mediation moderation analysis is a little confusing – why is depression the outcome variable in the one analysis? From what I had read, smartphone usage was the outcome variable. I found the results section confusing, and not always clear.
[Response]: Firstly, we would like to express our sincere apologies for any confusion caused and assure you that we value your comments and suggestions.
As we understand, a moderated mediation analysis is a process used to determine whether a moderating variable influences the strength of the relationship between the independent variable / mediating variable and the dependent variable. So we conducted this analysis to ascertain whether the independent variable affects the dependent variable via the mediating variable and moderating variable.
This typically involves two analyses: (a) Mediation Analysis: Analyze the mediating effect using appropriate statistical methods (regression analysis). In this step, the independent variable (perceived social support) is used to predict the mediating variable (depressive symptoms), and then they both are used to predict the dependent variable (mobile phone addiction). (b) Moderation Analysis: In this step, the moderation effect is inspected. We included an interaction term (perceived social support * self-compassion and depressive symptoms * self-compassion) in our model. This is to test whether the moderator modifies the relationship between the independent and the mediating variable or between the mediating variable and the dependent variable.
Thus, to clarify, the step - when the independent variable (perceived social support) is used to predict the mediating variable (depressive symptoms) - can be seen as an intermediate process in our analysis, rather than the final result. Mobile phone addiction is the outcome variable.
In practice, it’s not strictly necessary to first conduct a mediation analysis and then follow up with a moderated mediation analysis. Our initial idea was to use this method to provide a more comprehensive analysis. Unfortunately, it caused some confusion for you. We hope to have shed light on your query with this explanation. Thanks again for your valuable comment.
- The discussion starts of well explaining why depression may predict mobile phone use. However, the discussion surrounding the mediating and moderating effects is less clear and repetitive.
[Response]: We greatly appreciate your insightful feedback on our manuscript. In light of your helpful comments, we recognize that this part of our manuscript could indeed benefit from more precise explanation and less redundancy. Therefore, we have undertaken a careful review and revision of the discussion section for these effects. We hope that our revisions meet your expectations.
[Revision on page 12-15, Section 4]
- There are several places where the spelling and grammar is incorrect. I suggested a proofreader be employed to fix these errors as it often made the article hard to read.
[Response]: Thanks a lot for your supportive guidance and constructive feedback on our manuscript. Following your advice, we have meticulously checked and corrected the grammatical and spelling errors in our manuscript. Furthermore, to enhance the quality and readability of our paper, we have acted promptly and diligently by engaging a colleague fluent in English writing. We believe these amendments greatly improve the clarity and coherence of our work, and hope that the revised manuscript now meets the requirements of publication.
- I have left several comments in the attached PDF.
[Response]: We wish to express our heartfelt gratitude for the meticulous and valuable feedback provided in the PDF attached to your previous email. We have thoroughly examined your comments and have updated our manuscript accordingly.
We have also responded to the comments you left in the PDF one by one, and we have uploaded the attachments for your reference.
By addressing these concerns, we believe we have significantly advanced the clarity and rigor of our study. Your time, effort, and expertise in reviewing and suggesting improvements to our manuscript have been crucial to this process. Thanks again!

Reviewer 2 Report
Thank you for giving me the opportunity to referee this very interesting paper.
I have a few comments to make to improve the quality of the work:
- hypotheses should be separated from the theoretical framework
- the bibliography should be updated with references to the most recent literature
- the bibliography should be cited (I presume) taking into account the style proposed by Mdpi
- add a few more limitations
Moderate editing of English language required
Author Response
Thank you for giving me the opportunity to referee this very interesting paper.
I have a few comments to make to improve the quality of the work:
- hypotheses should be separated from the theoretical framework
[Response]: Thanks for your insightful suggestion. We have now reformulated the structure of our manuscript to clearly distinguish between the theoretical framework and the hypotheses. We have added section 1.4 to clarify our hypotheses more clearly, just as follows.
[1.4. The present study
The present study aims to examine the underlying mediator (depressive symptom) and moderator (self-compassion) between perceived social support and mobile phone addiction among college students. Specifically, this study constructed a moderated mediation model to test three main hypotheses:
Hypothesis 1. Depressive symptoms mediate the relationship between perceived social support and mobile phone addiction.
Hypothesis 2a. Self-compassion moderates the direct association. Perceived social support has a weaker association with mobile phone addiction among college students with a high level of self-compassion than those with a low level of self-compassion.
Hypothesis 2b. Self-compassion moderates the mediation processes. Perceived social support has weaker associations with depressive symptoms, which in turn have weaker associations with mobile phone addiction among college students with a high level of self-compassion compared to those with a low level of self-compassion.
The theoretical model is outlined in Fig. 1.]
[Revision on page 5, Section 1.4]
- the bibliography should be updated with references to the most recent literature
[Response]: Firstly, we would like to express our gratitude for your insightful suggestion to update our bibliography. Following your guidance, we have updated our reference list to incorporate the most recent literature. We have supplemented our bibliography with the latest studies related to our research, ensuring our analysis is grounded in the most current findings. We believe these amendments significantly enhance the vitality and relevance of our manuscript to present discourse in the field. We hope our modifications align with your suggestions.
The following are some of the updated references used for this study.
A, Y. P., B, H. Z., A, B. Z., A, H. M., A, R. H., & C, H. J. (2021). Perceived stress and mobile phone addiction among college students during the 2019 coronavirus disease: the mediating roles of rumination and the moderating role of self-control. Personality and Individual Differences. https://doi.org/10.1016/j.paid.2021.111222
Ding, Z. C., Yan, J., & Fu, J. (2021). Internet and mobile phone addiction self-control mediate physical exercise and subjective well-being in young adults using iot. Mobile information systems. https://doi.org/10.1155/2021/9923833
Hou, J., Zhu, Y., & Fang, X. (2021). Mobile phone addiction and depression: multiple mediating effects of social anxiety and attentional bias to negative emotional information. Acta Psychologica Sinica, 53(4), 362-373. https://doi.org/10.3724/SP.J.1041.2021.00362
Kong, F., Qin, J., Huang, B., Zhang, H., Lei, L., & Lindsey, D. (2020). The effect of social anxiety on mobile phone dependence among Chinese adolescents: A moderated mediation model. Children and Youth Services Review, https://doi.org/10.1016/j.childyouth.2019.104517
Lepp, A., Barkley, J. E., Sato, T., Yamatsu, K., & Glickman, E. (2020). Problematic smartphone use is negatively related to physical activity in american college students: 3471 board #292 may 29 1:30 pm - 3:00 pm. Medicine & Science in Sports & Exercise, 52. https://doi.org/10.1249/01.mss.0000686012.01230.e0
Qi. W. (2018). Middle school students' mobile phone dependence and its grade differences. Advances in Psychology, 08(6), 820-827. https://doi.org/10.12677/AP.2018.86098
Ran, G., Li, J., Zhang, Q., & Niu, X. (2022). The association between social anxiety and mobile phone addiction: a three-level meta-analysis. Computers in Human Behavior. https://doi.org/10.1016/j.chb.2022.107198
You, Z., Zhang, Y., Zhang, L., Xu, Y., & Chen, X. (2019). How does self-esteem affect mobile phone addiction? the mediating role of social anxiety and interpersonal sensitivity. Psychiatry Research, 271, 526-531. https://doi.org/10.1016/j.psychres.2018.12.040
- the bibliography should be cited (I presume) taking into account the style proposed by Mdpi
[Response]: We sincerely thank you for careful reading. We have since revised the manuscript and reformatted our references according to the style proposed by MDPI. We confirm that all citations within the text and the reference list now fully comply with the required format.
- add a few more limitations
[Response]: We sincerely appreciate your valuable comment. We have included some additional limitations in the revised manuscript and hope that they address your suggestions adequately.
The revised version of limitation is as follows.
[There are still limitations in this study, which can be improved and explored in future research. Firstly, this study is a cross-sectional study, so it must be cautious in inferring the causal relationships between these variables. Studies have found that mobile phone addiction can also lead to depression, anxiety, and stress [11, 54], so future studies could use longitudinal or experimental design to verify the causal relationship further.
Secondly, this study was based on self-reported data from online questionnaires. Research indicates that self-reported mobile phone usage does not consistently correlate strongly with objectively measured mobile phone use [55, 56]. Thus, future studies could adopt the multi-source method to collect data. What’s more, the gender proportion of participants is unbalanced. Although this study uses gender as a covariant in data analysis, it might affect the generalization of the results.
Thirdly, this study found that depressive symptoms could mediate perceived social support's effect on mobile phone addiction. However, whether other negative emotions (such like anxiety, etc) might have the same impact remains to be further explored.
Fourthly, our research focuses only on the moderating role of one positive psychological variable - self-compassion. However, many other positive psychological variables such as self-efficacy, optimism, and resilience may play a similar role. Future research might further investigate the interaction between these variables and their cumulative effects on individuals' responses to adverse situations.
Finally, breaking down the MPAI into its four dimensions (i.e., loss of control, withdrawal, avoidance, and inefficiency) could provide valuable insights into the specific aspects of mobile phone addiction that are most influenced by perceived social support, self-compassion and depressive symptoms. Considering this, we see a valuable direction for future research in carrying out a more elaborate separate dimension analysis of the MPAI. Besides,it is still unclear which functions of mobile phones (entertainment, information seeking, learning, and so on.) are related to the lack of social support, which is also worthy of exploration in future research.]
[Revision on page 14-15, Section 4.6]

Reviewer 3 Report
1. It is necessary to clarify the innovation of this article. Specifically, to what extent has the relevant research progressed, and where is the innovation of this research reflected?
2. Consider adding some latest literature.
3. The discussion part should not only explain the research results again, but also compare with previous similar studies. For example, have similar studies found the moderate effect of self-compassion in the past? What are the similarities and differences between these studies and this study?
4. What is the theoretical contribution of this study? Specifically, what theory does this research support or extend? Or any inspiration for what theory?
English writing quality is OK
Author Response
- It is necessary to clarify the innovation of this article. Specifically, to what extent has the relevant research progressed, and where is the innovation of this research reflected?
[Response]: We would like to express our gratitude for your insightful feedback.
Indeed, previous research on mobile phone addiction extensively examines the negative factors contributing to its development, with less emphasis on the buffering effect of positive variables. There are several studies that have addressed the impacts of positive variable on mobile phone addiction, such like self-control and self-esteemy [1,2]. However, in the realm of positive psychology, self-compassion has recently garnered significant attention and is defined as the capacity to offer empathy or compassion to oneself when confronted with failures, inadequacies, or distress [3].
Existing research has found a relationship between self-compassion and internet addiction [4]. And studies have also demonstrated that self-compassion moderates the association between peer victimization and mobile phone addiction among adolescents [5]. However, the issue is particularly prominent among university students, a high-risk group for mobile phone addiction. As a generation independently confronting societal pressures and academic stress with limited immediate parental care and assistance, university students often face the harsh challenge of insufficient social support.
Therefore, our research focuses on this demographic, exploring ways to alleviate the depressive symptoms brought on by a lack of social support, and the accompanying issue of mobile phone addiction. Self-compassion, as a positive psychological construct, has been recognized in academia for its protective role among vulnerable populations. With this in mind, we consider self-compassion as a moderator in our study, exerting its influence by moderating the impact of distal factors (like lack of social support) on proximal outcomes (like depressive symptoms leading to addiction). With the aim to investigate its role in alleviating mobile phone addiction among university students. This perspective is integral to understanding mobile phone addiction and developing effective interventions targeted towards students.
We believe our study provides a unique perspective on how to understand and address the issue of mobile phone addiction among university students, potentially contributing to the reduction of their psychological distress and over-reliance on mobile phones.
In light of your thoughtful feedback, we have strived to supplement our manuscript with a more detailed exposition underpinned by corroborative literature. We hope this response addresses your query. Once again, we are thankful for your valuable input.
[Revision on page 4, Section 1.3]
References
-
- A, Y. P., B, H. Z., A, B. Z., A, H. M., A, R. H., & C, H. J. (2021). Perceived stress and mobile phone addiction among college students during the 2019 coronavirus disease: the mediating roles of rumination and the moderating role of self-control. Personality and Individual Differences. https://doi.org/10.1016/j.paid.2021.111222
- You, Z., Zhang, Y., Zhang, L., Xu, Y., & Chen, X. (2019). How does self-esteem affect mobile phone addiction? the mediating role of social anxiety and interpersonal sensitivity. Psychiatry Research, 271, 526-531. https://doi.org/10.1016/j.psychres.2018.12.040
- Neff, K. (2003). Self-compassion: An alternative conceptualization of a healthy attitude toward oneself. Self and identity, 2(2), pp.85-101. https://doi.org/10.1080/15298860309032
- İSKENDER, Murat, & Akin, A. (2011). Self-compassion and internet addiction. Turkish Online Journal of Educational Technology, 10(3), 215-221. https://doi.org/10.1080/1475939X.2011.588414
- Liu, Q. Q., Yang, X. J., Hu, Y. T., & Zhang, C. Y. (2020). Peer victimization, self-compassion, gender and adolescent mobile phone addiction: unique and interactive effects. Children and Youth Services Review, 105397. https://doi.org/10.1016/j.childyouth.2020.105397
- Consider adding some latest literature.
[Response]: We sincerely appreciate your constructive comment. Following your recommendation, we have updated our reference list to include the latest research published in the past few years relevant to our study. We believe that this update provides a more comprehensive view of the current discourse on our research topic, and strengthens our analysis by supplementing it with the most recent empirical findings.
The following are some of the updated references used for this study.
A, Y. P., B, H. Z., A, B. Z., A, H. M., A, R. H., & C, H. J. (2021). Perceived stress and mobile phone addiction among college students during the 2019 coronavirus disease: the mediating roles of rumination and the moderating role of self-control. Personality and Individual Differences. https://doi.org/10.1016/j.paid.2021.111222
Ding, Z. C., Yan, J., & Fu, J. (2021). Internet and mobile phone addiction self-control mediate physical exercise and subjective well-being in young adults using iot. Mobile information systems. https://doi.org/10.1155/2021/9923833
Hou, J., Zhu, Y., & Fang, X. (2021). Mobile phone addiction and depression: multiple mediating effects of social anxiety and attentional bias to negative emotional information. Acta Psychologica Sinica, 53(4), 362-373. https://doi.org/10.3724/SP.J.1041.2021.00362
Kong, F., Qin, J., Huang, B., Zhang, H., Lei, L., & Lindsey, D. (2020). The effect of social anxiety on mobile phone dependence among Chinese adolescents: A moderated mediation model. Children and Youth Services Review, https://doi.org/10.1016/j.childyouth.2019.104517
Lepp, A., Barkley, J. E., Sato, T., Yamatsu, K., & Glickman, E. (2020). Problematic smartphone use is negatively related to physical activity in american college students: 3471 board #292 may 29 1:30 pm - 3:00 pm. Medicine & Science in Sports & Exercise, 52. https://doi.org/10.1249/01.mss.0000686012.01230.e0
Qi. W. (2018). Middle school students' mobile phone dependence and its grade differences. Advances in Psychology, 08(6), 820-827. https://doi.org/10.12677/AP.2018.86098
Ran, G., Li, J., Zhang, Q., & Niu, X. (2022). The association between social anxiety and mobile phone addiction: a three-level meta-analysis. Computers in Human Behavior. https://doi.org/10.1016/j.chb.2022.107198
You, Z., Zhang, Y., Zhang, L., Xu, Y., & Chen, X. (2019). How does self-esteem affect mobile phone addiction? the mediating role of social anxiety and interpersonal sensitivity. Psychiatry Research, 271, 526-531. https://doi.org/10.1016/j.psychres.2018.12.040
- The discussion part should not only explain the research results again, but also compare with previous similar studies. For example, have similar studies found the moderate effect of self-compassion in the past? What are the similarities and differences between these studies and this study?
[Response]: Thank you for your valuable feedback. We greatly value your feedback and have taken your comments into consideration while revising our discussion section. The discussion section now contains additional comparative analysis with previous similar studies, addressing their similarities and differences to our findings, as per your suggestion. The changes highlight the unique nature of our study - exploring the moderating effect of self-compassion in the context of perceived social support, depressive symptoms, and mobile phone addiction among university students, and significantly expands upon the implications of our results within the broader research context.
[Previous studies have explored the moderating role of self-compassion in various contexts, such as its impact on the relationship between stress and relevant behavioral outcomes [41, 42]. While these studies have demonstrated the protective effects of self-compassion on adverse psychological consequences, they have rarely focused on mobile phone addiction, particularly in a university student population. Our study is the first to explore the moderating role of self-compassion in the relationship between perceived social support and mobile phone addiction. The findings suggest that inadequate perceived social support increased the risk of mobile phone addiction among college students due to the role of depressive symptoms. Nevertheless, self-compassion could buffer this adverse effect. ]
[Revision on page 11, Section 4]
- What is the theoretical contribution of this study? Specifically, what theory does this research support or extend? Or any inspiration for what theory?
[Response]: Thanks a lot for your supportive guidance and constructive feedback on our manuscript.
Firstly, our research broadens the understanding of mobile phone addiction among college students. We've analyzed this within an integrated framework incorporating perceived social support, depressive symptoms, and self-compassion. Consequently, our study not only supports, but also extends on the Compensatory Internet Use theory[1] and the Interpersonal Model [2].
Secondly, our research introduces the concept of self-compassion as a moderation variable in this relationship, which is a relatively newer area in the realm of positive psychology pertaining to addictive behaviors. Our findings underscore the potential of positive psychological constructs (self-compassion), as instrumental in driving positive behavior change even in adverse situations. This cultivates a more nuanced understanding of protective factors in the onset of phone addiction and aligns with the principles of emotion-oriented coping strategy [3].
Finally, in practice, our research sparks reconsideration for intervention strategies addressing addiction. Moving from traditional external support systems, it presents the promise of harnessing internal resourcefulness through the cultivation of self-compassion. This inspiration can potentially develop into a new theoretical framework for addressing addiction and similar maladaptive behavior patterns.
We have incorporated this explanation into the revised manuscript. We hope that this adequately addresses your question.
[Revision on page 13, Section 4.4]
References
-
- Kardefelt-Winther, & Daniel. (2014). A conceptual and methodological critique of internet addiction research: towards a model of compensatory internet use. Computers in Human Behavior, 31(31), 351-354.
- Cohen, S., & Mckay, G. (1984). Social support, stress, and the buffering hypothesis: A theoretical analysis. Hillsdale.
- Stanton, A.L., Kirk, S.B., Cameron, C.L. & Danoff-Burg, S. (2000). Coping through emotional approach: scale construction and validation. Journal of personality and social psychology, 78(6), p.1150. https://doi.org/10.1037/0022-3514.78.6.1150

Reviewer 4 Report
Thanks for the opportunity for review this article. This article conduct a cross-sectional study on relationship between self-compassion, depressive symptoms, mobile phone addiction and perceived social support. This study provides some valuable information. I have suggested authors to clarify the theoretical background of the study. As a cross-sectional study, the order of the variables in the model are rather interchangeable, and thus more theoretical supports or arguments are needed. Furthermore, authors should highlights their innovative contributions. Please see detailed suggestions below.
Major suggestions:
While H1 is reasonable, please consider whether all of these hypotheses are key hypotheses that are focused in the current study. The point is, key hypotheses should link to the innovative exploration, and very basic findings could skip in hypothesis.
The relationship between depression, social support and mobile phone addiction require a further clarification that reject other order in the mediation (e.g., who not depression as dependent variable?)
I'm quite concerned with the role of self-compassion. I think the role of moderator should be further illustrated. "buffer against depressive symptom caused by environmental stress" is not enough for supporting a moderator role I think. More importantly, the authors hypothesized self-compassion serve as moderators on all three paths? I think these need some arguments.
While the hypotheses sound reasonable, the authors should highlight how the current study is innovative or contributive to the field.
Method
DASS three dimensions were measured but only depression was used? No matter what is the fact, please clarify this. Especially, authors mentioned potential contribution of anxiety in the limitation section.
Results
Did authors consider to further explore the four dimensions of MPAI?
Why did authors divide male and female? Is this demographic variable important to this study? If so, whether authors want to divide model according to gender?
Discussion
Consistent with introduction, I suggest authors to highlight the innovative contribution, or interesting findings of the results.
Note that self-compassion were not significant at all locations of moderation. But it seems the discussion did not specify the details, which leaving an impression that specific locations are not important at all?
Conclusion
In my opinion, conclusion should be a take home message about the finding or the generalized fact (e.g., depression mediate social support and mobile addiction), rather than say this study explored relationship between a, b, c.
Minor suggestions:
The first paragraph of introduction should have some citations.
Please forgive me if I miss it: what is the number of participants for M+1SD, M-1SD for self-compassion? I did not find this.
Author Response
- Thanks for the opportunity for review this article. This article conduct a cross-sectional study on relationship between self-compassion, depressive symptoms, mobile phone addiction and perceived social support. This study provides some valuable information. I have suggested authors to clarify the theoretical background of the study. As a cross-sectional study, the order of the variables in the model are rather interchangeable, and thus more theoretical supports or arguments are needed. Furthermore, authors should highlights their innovative contributions. Please see detailed suggestions below.
[Response]: We sincerely appreciate the time and effort you dedicated to reviewing our manuscript. Your constructive feedback has provided us with valuable insights to refine our research. We have taken all of your detailed suggestions into account during the revision process, and we believe the manuscript now presents a clearer, theoretically stronger, and more compelling version of our study. We're grateful for your contributions to this improvement.
Major suggestions:
- While H1 is reasonable, please consider whether all of these hypotheses are key hypotheses that are focused in the current study. The point is, key hypotheses should link to the innovative exploration, and very basic findings could skip in hypothesis.
[Response]: Thank you for your insightful comment. We agree that our hypotheses should be more directly aligned with the novel aspects of our study. And we have added section 1.4 to clarify our hypotheses more clearly, just as follows. We have incorporated these changes into the revised manuscript and hope this adequately addresses your feedback.
[1.4. The present study
The present study aims to examine the underlying mediator (depressive symptoms) and moderator (self-compassion) between perceived social support and mobile phone addiction among college students. Specifically, this study constructed a moderated mediation model to test three main hypotheses:
Hypothesis 1. Depressive symptoms mediate the relationship between perceived social support and mobile phone addiction.
Hypothesis 2a. Self-compassion moderates the direct association. Perceived social support has a weaker association with mobile phone addiction among college students with a high level of self-compassion than those with a low level of self-compassion.
Hypothesis 2b. Self-compassion moderates the mediation processes. Perceived social support has weaker associations with depressive symptoms, which in turn have weaker associations with mobile phone addiction among college students with a high level of self-compassion compared to those with a low level of self-compassion.
The theoretical model is outlined in Fig. 1. ]
[Revision on page 5, Section 1.4]
- The relationship between depression, social support and mobile phone addiction require a further clarification that reject other order in the mediation (e.g., who not depression as dependent variable?)
[Response]: Thank you for constructive comment.
To support the arrangement of our variables, we lean on cognitive-behavioral theory, which posits that maladaptive behaviors (like addiction) often result from impaired cognitive processes, offering a theoretical basis for treating depressive symptom as a proximal factor leading to mobile phone addiction[1].
According to cognitive-behavioral theory, individuals with depressive symptoms may engage in maladaptive behaviors like excessive mobile phone use as an attempt to alleviate or escape from their negative emotions. In this sense, depressive symptoms can serve as a key psychological mechanism or "proximal factor" that triggers the addictive behavior.
Lastly, another theoretical basis for this study (and our handling of the variables) is grounded in the theoretical cognitive model of substance abuse by Beck et al. (1993) [2]. According to this model, individuals who are unable to interpret negative emotions correctly are more prone to addictive behaviors as they often turn to unhealthy coping mechanisms, such as substance use (or, in our case, excessive mobile phone use) to manage their negative emotions.
In the context of our study, if individuals misinterpret their depressive symptoms, they might excessively use mobile phones as an improper coping strategy – hence the mobile phone addiction.
[1] Wu-Ouyang, B. (2022). Are smartphones addictive? examining the cognitive-behavior model of motivation, leisure boredom, extended self, and fear of missing out on possible smartphone addiction. Telematics and informatics. https://doi.org/10.1016/j.tele.2022.101834
[2] Beck, A. T., Wright, F. D., Newman, C. F., & Liese, B. S. (1993). Cognitive therapy of substance abuse. New York, NY: Guilford Press.
[Revision on page 4, Section 1.2]
- I'm quite concerned with the role of self-compassion. I think the role of moderator should be further illustrated. "buffer against depressive symptom caused by environmental stress" is not enough for supporting a moderator role I think. More importantly, the authors hypothesized self-compassion serve as moderators on all three paths? I think these need some arguments.
[Response]: We appreciate your insightful remarks regarding the role of self-compassion in our research.
In our study, self-compassion has been proposed as a moderating variable based on the theoretical premises that it is a self-regulatory strategy that can function effectively across varying emotional conditions. The premise is that individuals with higher levels of self-compassion may be more resilient in dealing with the impacts of environmental stressors and depressive symptoms, thereby diminishing the tendency toward addictive behaviors such as mobile phone addiction.
Specifically, we suggest that self-compassion moderates the relationship between environmental stress (e.g., lack of social support) and depressive symptoms, environmental stress (e.g., lack of social support) and mobile phone addiction, as well as the relationship between depressive symptoms and mobile phone addiction.
Based on our study findings, self-compassion did not significantly moderate the relationship from depressive symptoms to mobile phone addiction. This could be due to several reasons. Firstly, while self-compassion is generally considered a significant factor in mitigating the effects of negative emotions, its impact may not be as pronounced within the specific connection between depressive symptoms and addictive mobile phone behavior.
Secondly, this finding might also be linked to the characteristics of our sample. For instance, if many individuals within our sample had relatively high levels of self-compassion, then this factor may not provide significant moderation in the relationship between depressive symptoms and mobile phone addiction.
Lastly, there may exist other variables that we have not taken into account which could obscure the moderating effect of self-compassion in the link between depressive symptoms and mobile phone addiction.
Regardless, the issue you've pointed out is one that requires further investigation for a better understanding of the exact role of self-compassion in this process, and under what conditions its impact might become apparent or diminish. This finding also underscores the need for future research to delve deeper into the role of self-compassion and other potential factors that might influence its functionality.
Hope the answer helps clarify our findings. Once again, we want to express our gratitude for your constructive feedback. It significantly contributes to enhancing the quality and clarity of our study.
- While the hypotheses sound reasonable, the authors should highlight how the current study is innovative or contributive to the field.
[Response]: Thank you for your feedback. We recognize the importance of articulating how our study is innovative and contributive to the field. To address this, we highlighted the following aspects in our revised manuscript:
Elucidating the Mediating Role of Depressive Symptoms: The identification of depressive symptoms as a mediator in the relationship between perceived social support and mobile phone addiction is a significant contribution of our research. This amplifies our comprehension of the potential role of emotional well-being in influencing addictive behaviors.
Highlighting the Role of Self-Compassion: The exploration of self-compassion as a moderating variable in the context of mobile phone addiction is relatively novel in this research domain. Our findings suggest that increasing self-compassion can buffer the adverse impacts of low social support and depressive symptoms, thereby potentially preventing or reducing mobile phone addiction. This opens avenues for future preventative and intervention strategies.
We appreciate your constructive feedback as it helps us improve the quality of our work.
Method
- DASS three dimensions were measured but only depression was used? No matter what is the fact, please clarify this. Especially, authors mentioned potential contribution of anxiety in the limitation section.
[Response]: Thanks for your detailed comment.
The reason for focusing on depressive symptoms, specifically, was based on previous research demonstrating a connection between depression and mobile phone addiction. We hypothesized that depressive symptoms may act as a mediator in the relationship between perceived social support and mobile phone addiction, which was the central focus of our investigation.
However, we acknowledge in the limitation section the potential contributions of anxiety and stress within this context. Indeed, past research suggests that these factors might also play significant roles in the domain of mobile phone addiction. Although these were not the primary focus of this study, we recognize the potential contribution of examining these as well in future research for a more comprehensive understanding of addictive mobile phone behaviors.
We have updated our manuscript to further clarify why we focused on depressive symptoms. We greatly appreciate your suggestion and will consider incorporating the other dimensions of the DASS for more holistic future studies.
[Revision on page 6, Section 2.2.3]
Results
- Did authors consider to further explore the four dimensions of MPAI?
[Response]: We appreciate your suggestion to delve deeper into the four dimensions of the Mobile Phone Addiction Index (MPAI). Indeed, breaking down the MPAI into its four dimensions (i.e., loss of control, withdrawal, avoidance, and inefficiency) could provide valuable insights into the specific aspects of mobile phone addiction that are most influenced by perceived social support, self-compassion and depressive symptoms.
However, in our current research, we have chosen to focus on the overall phenomenon of mobile phone addiction. This was done in order to provide a more generalizable understanding of how perceived social support, self-compassion and depressive symptoms can influence addictive tendencies.
Nevertheless, we acknowledge the potential benefits of a dimension-wise analysis, which could potentially uncover more nuanced and detailed information about the various factors contributing to specific facets of mobile phone addiction. Considering this, we see a valuable direction for future research in carrying out a more elaborate separate dimension analysis of the MPAI.
We thank you once again for your suggestion and have incorporated this idea into the discussion section of our revised manuscript as a potential avenue for future research.
[Revision on page 17, Section 4.6]
- Why did authors divide male and female? Is this demographic variable important to this study? If so, whether authors want to divide model according to gender?
[Response]: Thank you for your insightful question regarding the consideration of gender in our study.
Indeed, the decision to divide participants into male and female groups was rooted in previous research which suggests that gender can play a significant role in mobile phone addiction [1]. In general, males and females may differ in their coping strategies, emotional responses, and levels of self-compassion, therefore we believe it is crucial to acknowledge these potential differences within our study.
However, while we considered gender in the demographic analysis, we did not form separate models based on it in our primary exploration. This is mainly because our research focus is on the mediating role of depressive symptoms and moderating role of self-compassion and not primarily on gender differences.
Furthermore, in order to avoid any potential confusion and to focus on our main objective, we have decided to remove the t-test of gender. However, we have retained gender, along with age and grade, as control variables in our model, because of their potential influence on mobile phone addiction as suggested by the existing literature [1-3].
-
- Jenaro, C., Flores, N., Gómez-Vela, María, González-Gil, Francisca, & Caballo, C. (2007). Problematic internet and cell-phone use: psychological, behavioral, and health correlates. Addiction Research & Theory, 15(3), 309-320.
- Beranuy, M., Oberst, U., Carbonell, X., & Chamarro, A. (2009). Problematic Internet and mobile phone use and clinical symptoms in college students: The role of emotional intelligence. Computers in Human Behavior. 25(5). 1182-1187.
- Bian, M., & Leung, L. (2014). Smartphone Addiction: Linking Loneliness, Shyness, Symptoms and Patterns of Use to Social Capital. Media Asia. 41(2). 159-176.
[Revision on page 8, Section 3.2]
Discussion
- Consistent with introduction, I suggest authors to highlight the innovative contribution, or interesting findings of the results.
[Response]: Thank you for your feedback, and we fully appreciate the importance of highlighting the innovative elements of our study. In our revised conclusion, we have emphasized the following groundbreaking findings:
Clarifying the Mediating Role of Depressive Symptoms: Our research contributes to the field by identifying depressive symptoms as a mediator between perceived social support and mobile phone addiction. This is significant in understanding the potential role of emotional health in influencing addictive behaviors.
Emphasizing the Role of Self-Compassion: Our study introduces self-compassion as a moderating variable in examining mobile phone addiction, which is relatively new in this field. Our findings suggest that enhancing self-compassion may alleviate the negative impact of low social support and depression on mobile phone addiction, thereby opening new avenues for future prevention and intervention strategies.
[Revision on page 11-15, Section 4]
- Note that self-compassion were not significant at all locations of moderation. But it seems the discussion did not specify the details, which leaving an impression that specific locations are not important at all?
[Response]: Thank you for raising this important question. Thank you for your insightful comment, which prompted us to revisit the scope of our discussion.
In response to your comment, we would like to further elaborate: While our results indicate a non-significant moderating effect of self-compassion, this does not necessarily imply its insignificance in the broader context of mental health and behavior. As a psychological trait, self-compassion might play a more preventative role by possibly reducing the onset of depressive symptoms rather than mitigating the behavior (excessive mobile phone usage) that may occur once depressive symptoms have manifested.
To put it differently, individuals with high levels of self-compassion might generally be less prone to develop depressive symptoms, which in turn reduces the possibility of engaging in addictive mobile phone behavior. Subsequent to your comment, we have realized that these important nuances were not articulated as clearly as they should have been in our initial discussion.
We appreciate your observation and we will amend our discussion to provide a more comprehensive interpretation of our findings.
[Revision on page 13, Section 4.3]
Conclusion
- In my opinion, conclusion should be a take home message about the finding or the generalized fact (e.g., depression mediate social support and mobile addiction), rather than say this study explored relationship between a, b, c.
[Response]: We appreciate your valuable feedback. Based on your input, we will revise our conclusion to emphasize our primary findings. This revision encapsulates our central discoveries, providing a clear take-home message about the finding.
[Revision on page 11-15, Section 4]
Minor suggestions:
- The first paragraph of introduction should have some citations.
[Response]: We appreciate your valuable comment. We have revised our introduction to include references that support our study. Thank you for pointing out this important detail. We highly appreciate your deep understanding and careful consideration of academic research norms which contribute to the improvement of our paper.
[1] Ran, G., Li, J., Zhang, Q., & Niu, X. (2022). The association between social anxiety and mobile phone addiction: a three-level meta-analysis. Computers in Human Behavior, 130, 107198-.https://doi.org/10.1016/j.chb.2022.107198
[2] Lepp, A., Barkley, J. E., Sato, T., Yamatsu, K., & Glickman, E. (2020). Problematic smartphone use is negatively related to physical activity in american college students: 3471 board #292 may 29 1:30 pm - 3:00 pm. Medicine & Science in Sports & Exercise, 52. https://doi.org/10.1249/01.mss.0000686012.01230.e0
[3] Ding, Z. C., Yan, J., & Fu, J. (2021). Internet and mobile phone addiction self-control mediate physical exercise and subjective well-being in young adults using iot. Mobile information systems. https://doi.org/10.1155/2021/9923833
[Revision on page 1-2, Section 1]
- Please forgive me if I miss it: what is the number of participants for M+1SD, M-1SD for self-compassion? I did not find this.
[Response]: We appreciate your meticulousness in the examination of our study, and allow us to clarify your concerns.
In moderated mediation analysis, utilizing ±1 Standard Deviation (SD) from the Mean (M) aims to illustrate the differences in scores between individuals with high and low levels of self-compassion. Such an approach helps to determine whether the moderating effect is significant. It is not standard practice to report the number of participants falling within ±1SD [1]. However, understanding the relevance of this information to inter-study comparability and replicability, we will follow your advice.
To heed your suggestion, we have included the count of participants falling within M±1SD for self-compassion in our revised manuscript. In our sample size of (874), 227 fell within M+1SD and 443 within M-1SD for the self-compassion measure.
We greatly value your feedback, and we believe this additional detail substantially enhances both the completeness and the interpretability of our analyses.
[1]A, Y. P., B, H. Z., A, B. Z., A, H. M., A, R. H., & C, H. J. (2021). Perceived stress and mobile phone addiction among college students during the 2019 coronavirus disease: the mediating roles of rumination and the moderating role of self-control. Personality and Individual Differences. https://doi.org/10.1016/j.paid.2021.111222
[Revision on page 9-10, Section 3.3]

Round 2
Reviewer 1 Report
Thank you for taking the time to make the suggest edits.
I am happy with the revised manuscript and endorse this manuscript for publication.
The minor typos have been vastly improved.
Reviewer 4 Report
Thanks for the reply and corresponding revision. I have no more suggestions, but please take care that some of your references have formatting issues, please consider to check them in the next step of publishing.